# A computational approach to identify phytochemicals as potential inhibitor of acetylcholinesterase: Molecular docking, ADME profiling and molecular dynamics simulations

**Mahir Azmal[1], Md. Sahadot Hossen[1], Md. Naimul Haque Shohan[1], Rashid Taqui[1], Abbeha Malik[2]\*, Ajit Ghosh[1]\***

1 Department of Biochemistry and Molecular Biology, Shahjalal University of Science and Technology, Sylhet, Bangladesh, 2 Department of Bioinformatics, Institute of Biochemistry, Biotechnology and Bioinformatics, The Islamia University of Bahawalpur, Punjab, Pakistan

\* abbeha.malik@iub.edu.pk (AM); aghosh-bmb@sust.edu (AG)

**Data Availability Statement:** All relevant data are within the paper and its Supporting information files.

## Abstract

Inhibition of acetylcholinesterase (AChE) is a crucial target in the treatment of Alzheimer's disease (AD). Common anti-acetylcholinesterase drugs such as Galantamine, Rivastigmine, Donepezil, and Tacrine have significant inhibition potential. Due to side effects and safety concerns, we aimed to investigate a wide range of phytochemicals and structural analogues of these compounds. Compounds similar to the established drugs, and phytochemicals were investigated as potential inhibitors for AChE in treating AD. A total of 2,270 compound libraries were generated for further analysis. Initial virtual screening was performed using Pyrx software, resulting in 638 molecules showing higher binding affinities compared to positive controls Tacrine (-9.0 kcal/mol), Donepezil (-7.3 kcal/mol), Galantamine (-8.3 kcal/mol), and Rivastigmine (-6.4 kcal/mol). Subsequently, ADME properties were assessed, including blood-brain barrier permeability and Lipinski's rule of five violations, leading to 88 compounds passing the ADME analysis. Among the rivastigmine analogous, [3-(1-methylpiperidin-2-yl)phenyl] N,N-diethylcarbamate showed interaction with Tyr123, Tyr336, Tyr340, Phe337, Trp285 residues of AChE. Tacrine similar compounds, such as 4-amino-2-styrylquinoline, exhibited bindings with Tyr123, Phe337, Tyr336, Trp285, Trp85, Gly119, and Gly120 residues. A phytocompound (bisdemethoxycurcumin) showed interaction with Trp285, Tyr340, Trp85, Tyr71, and His446 residues of AChE with favourable binding. These findings underscore the potential of these compounds as novel inhibitors of AChE, offering insights into alternative therapeutic avenues for AD. A 100ns simulation analysis confirmed the stability of protein-ligand complex based on the RMSD, RMSF, ligand properties, PCA, DCCM and MMGBS parameters. The investigation suggested 3 ligands as a potent inhibitor of AChE which are [3-(1-methylpiperidin-2-yl)phenyl] N,N-diethylcarbamate, 4-Amino-2-styrylquinoline and bisdemethoxycurcumin. Furthermore, investigation, including *in-vitro* and *in-vivo* studies, is

**Funding:** The author(s) received no specific funding for this work.

**Competing interests:** The authors have declared that no competing interests exist.

needed to validate the efficacy, safety profiles, and therapeutic potential of these compounds for AD treatment.

## Introduction

Alzheimer's disease (AD) is a neurological disorder that leads to the deterioration of brain cells. It is the primary cause of dementia, a condition marked by a decline in cognitive abilities and a loss of independence in daily tasks [1]. AD is characterized by a decline in the cholinergic system, resulting in reduced levels of acetylcholine in brain regions responsible for learning, memory, behavior, and emotional responses [2]. AD is neuropathologically defined by the presence of beta-amyloid (Aβ) plaques, neurofibrillary tangles, and degeneration or atrophy of the basal forebrain cholinergic neurons [3].

Acetylcholinesterase (AChE), an enzyme that belongs to the serine hydrolase family, plays a vital role in breaking down acetylcholine (ACh) into choline and acetate. Therefore, maintaining normal cholinergic neurotransmission. In AD patients ACh degradation is amplified by the AChE in early stages. The use of enzymatic inhibition to reduce AChE activity has shown promise as a treatment strategy for AD [4]. The FDA-approved AChE enzyme inhibitors donepezil and rivastigmine are utilized for the treatment of mild to moderate AD. Tacrine was one of the AChE inhibitory drugs which had been banned since 2013. Tacrine have adverse effects such as nausea, diarrhoea, loss of appetite, fainting, abdominal pain, and vomiting [5]. Administration of tacrine for AD treatment leads to reversible hepatotoxicity in 30–50% of patients, as evidenced by an elevation in transaminase levels [6]. Hence, researchers aim to identify novel therapeutics characterized by heightened efficacy and reduced incidence of adverse reactions [7].

Researchers have investigated natural resources for anti-AChE agents because they are safer than synthetic chemicals [8]. Galantamine, a natural drug from *Galanthus woronowii*, is used to treat AD alongside other chemical drugs [9]. However, none of these medications have proven to be entirely effective in halting the advancement or formation of AD. To ameliorate the potential side effects and optimize the therapeutic efficacy of enzyme inhibition, compounds possessing structural similarities to FDA-approved drugs emerge as promising candidates [10–12]. Ongoing research is being conducted to discover novel compounds derived from natural sources or FDA-approved drug-like compounds with anti-AChE properties [13]. Plant products and its derivatives are increasingly being recognized globally for their potential as AChE inhibitors (AChEi), making them a promising therapeutic option for the treatment of AD [14]. Extensive research has identified a comprehensive list of plant-derived substances that inhibit AChE. The research on AChE inhibition-based treatment of AD has focused on this diverse range of phytochemicals due to the absence of promising, effective, and safe inhibitors [8, 15].

Studies have demonstrated that memory-enhancing herbs such as *Enhydra fluctuans*, *Vanda roxburghii*, *Bacopa monnieri*, *Centella asiatica*, *Convolvulus pluricaulis*, and *Aegle marmelos* have AchE inhibitory and antioxidant properties [16]. This study aims to elucidate the human AChE inhibitory potential of the current FDA-approved drugs like structural analogues, as well as phytochemicals. Our study aimed to assess the *in-silico* assay results by employing various techniques such as molecular docking, ADME, MD simulation (RMSD, RMSF, and ligand properties). Principal component analysis (PCA) and Domain cross-correlation matrix (DCCM) analysis were performed to identify the main directions of motion of protein during the attachment of ligands throughout the simulation. Finally, all the analyses

were compared with respect to the FDA-approved drugs (donepezil, galantamine, and rivastigmine). Moreover, molecular mechanics with generalised born and surface area solvation (MM/GBSA) was performed to check the interaction energies of all categories such as H-bond, lipophilicity etc.

## Materials and methods

### Ligand selection

**Ligand library 1: Similar structure selection.** The rationale behind constructing library 1 (Similar structure search) was two-sided. Firstly, compounds with analogous structures might be able to show a similar kind of effect to some extent. Secondly, studies have reported mild to severe adverse effects upon their administration and among them. Each of the four compounds was used as a query in the PubChem database followed by a similar structure search.

**Ligand library 2: Dr. Duke database search for phytochemicals.** Phytochemicals, known for their anti-AChE and anti-butyrylcholinesterase (BChE) activities, were identified through a literature review of medicinal plants. Scientific names were queried in Dr. Duke's Phytochemical and Ethnobotanical Databases (https://phytochem.nal.usda.gov/). Compound names were then searched in PubChem for 3-D structure retrieval.

### Selection of target protein and protein preparation

The RSCB-PDB database (https://www.rcsb.org) was utilized to search for the target protein, human acetylcholinesterase protein (PDB ID: 4M0E) with a lower X-ray resolution (2.00 AÅ). Several gaps were spotted while checking the structure with PyMol. Both the docking and simulation processes were vulnerable to interference from missing residues. To avoid any subsequent anomaly in docking and molecular dynamics simulation the spotted missing residues were repaired. To ensure the missing residues I-tasser (https://zhanggroup.org/I-TASSER/) a web-based server was used to predict the 3D structure of protein. The FASTA sequence was retrieved from the RCSB PDB database and used to build the predicted structure. The geometry analysis was performed using the MolProbity server (http://molprobity.biochem.duke.edu/), and the overall geometry and Ramachandran plots were analyzed.

### Active site prediction

The active region on the surface of the protein that performs protein function is known as a protein-ligand binding site. To avoid blind docking the specific amino acid residues (S1 Table) of protein-ligand interaction were predicted using CASTP v3.0 (http://sts.bioe.uic.edu/castp/calculation.html).

### Molecular docking of primarily selected molecules

PyRx 0.8 was used for the initial virtual screening [17]. The protein was retrieved from the I-tasser website in PDB format after homology modelling and ligands were downloaded from the PubChem of NCBI (https://pubchem.ncbi.nlm.nih.gov) one by one in SDF file format.

The target protein was loaded in Pyrx 0.8 and converted into macromolecules. The similar structures of tacrine, donepezil, rivastigmine and galantamine along with phytochemicals were loaded in the PyRx virtual screening tool in triplicates. After energy minimization, it was converted into a pdbqt file. All the parameters and grid box positioned at some standard value (Centre box: X = -0.9600, Y = -38.1677, Z = 34.2085) and the dimensions in Angstrom were X = 58.7652, Y = 60.0782 and Z = 65.867. Later, the docking results were screened for binding affinity and then all the generated possible docked conformations were stored in CSV format

[17]. Only those conformations that interacted specifically with the active-site residues of the target protein targeted protein were selected and further detailed interactions were explored through Discovery Studio and PyMOL.

Re-docking was performed by the AutoDock Vina tool and HDOCK (http://hdock.phys. hust.edu.cn/) [18] for the reliability of the software, and consistency of the docking algorithm. The target protein was converted into pdbqt. The parameters and grid box were positioned at some standard value (Centre box: X = 106.848, Y = 43.703, Z = 18.797) and the dimensions of Box in Angstrom were X = 126, Y = 116 and Z = 122. Docking results of triplicates were reported as mean ± standard deviation as a negative value in kcal/mol where the lowest docking score indicates the highest binding affinity [19]. For Hdock docking the scores were compared with the control and ligands.

## ADME profiling

The SwissADME (http://www.swissadme.ch/index.php) server was utilized to conduct ADME profiling. Canonical smiles of ligands were required for conducting ADME analysis. To perform ADME profiling, the canonical smiles of all the ligands were uploaded as input on the SwissADME server. The entirety of the data was acquired in the CSV (comma-separated value) format. The subsequent sorting procedure was conducted according to the permeability of the blood-brain barrier, greater binding affinity, violations of drug-likeness violation (Lipinski, Ghose, Veber, Egan, Muggue), and oral bioactivity (lipophilicity, flexibility, solubility, instability, size) [20].

## Molecular dynamics simulation

Protein-ligand interaction stability during macromolecule structure-to-function transitions was studied using molecular dynamics. The Desmond software, developed by Schrödinger LLC, enabled the execution of molecular dynamics (MD) simulations that lasted for a duration of 100 nanoseconds. The simulations, utilizing Newton's classical equation of motion, monitored the path of atoms as they moved through time. The receptor-ligand complex was subjected to preprocessing using Maestro's Protein Preparation Wizard, which included optimization and minimization procedures. The system was prepared using the System Builder tool, employing the Transferable Intermolecular Interaction Potential 3 Points (TIP3P) solvent model within an orthorhombic box. The simulation was governed by the OPLS 2005 force field, and counter ions were introduced to maintain model neutrality. A 0.15 M sodium chloride (NaCl) solution was added to replicate the conditions found in the body. The simulations were conducted using the Number of particles (N), Pressure (P), and Temperature (NPT) ensemble, with a temperature of 300 K and a pressure of 1 atm. Before the simulation, the models underwent a process of relaxation. The trajectories were recorded at intervals of 100 picoseconds. The stability was evaluated by comparing the root mean square deviation (RMSD), root mean square fluctuation (RMSF), and ligand properties (radius of Gyration, Molecular surface area, hydrogen bond etc.). Analysis of PCA and DCCM were performed using Desmond software with default parameters [21, 22]. Additionally, a subsequent 100 ns simulation was conducted to further validate the findings, with MM/GBSA utilized to assess the binding stability over time and identify the optimal binding configuration.

Re-simulation for further validation of the data is performed by Gromacs simulation Software conserving the parameters unchanged. The stability was evaluated by comparing the root mean square deviation (RMSD), root mean square fluctuation (RMSF), and protein-ligand properties (radius of Gyration, SASA etc.).

**Table 1. Primary selection criteria for similar structure compounds.**

| Compound name/Criteria | Molecular Weight G/MOL [Min-Max] | Rotatable Bond Count [Min-Max] | Heavy Atom Count [Min-Max] | H-Bond Donor Count [Min-Max] | H-Bond Acceptor Count [Min-Max] | Polar Area, [Angstrom sq] [Min-Max] | Complexity [Min-Max] | XLOGP [Min-Max] |
|---|---|---|---|---|---|---|---|---|
| Tacrine | 147–467 | 0–9 | 12–30 | 0–4 | 0–10 | 4.9–104 | 144–494 | 1–5 |
| Donepezil | 289–479 | 4–9 | 21–35 | 0–2 | 2–8 | 26.3–119 | 366–776 | 2–5 |
| Rivastigmine | 179.26–479 | 2–9 | 13–30 | 0–3 | 2–7 | 12.2–112 | 147–497 | -0.3–4.7 |
| Galantamine | 245–445 | 0–9 | 18–32 | 0–4 | 2–9 | 18.5–128 | 326–766 | -2.4–4 |

## Results

### Ligand library construction

The number of similar structure compounds was massive; however, considering the facts about drug-likeness several criteria were optimized to select the best-suited structures. A total of 2252 similar compounds (library 1) and 18 phytochemicals (library 2) were primarily selected for the virtual screening based on the selection criteria (Table 1).

### 3D structure prediction

The I-tasser gave a modelled structure (Fig 1B) which is like the pdb 4M0E structure (Fig 1A), and mostly conserved (Fig 1C). The alignment of the sequence of amino acids is provided to verify the residues, with further sequence alignment and geometry details (S2 and S3 Tables). The Ramachandran plot (S1 Fig.) shows the statistical distribution of the combinations of the backbone dihedral angles ϕ and ψ. In theory, the allowed regions of the Ramachandran plot show which values of the Phi/Psi angles are possible for an amino acid, X, in an ala-X-ala tri-peptide [23]. The Ramachandran plot analysis of protein AChE showed high conformational quality, with no outliers identified. All 537 residues (100%) were in acceptable regions (>99.8%), with 96.6% (519/537) falling within favoured regions (>98%). The findings show the strong structural integrity of AChE [24].

### Virtual screening with PyRx

Using PyRx 0.8 docking tools, the original phytochemicals, and four other groups of similar structure were docked. The affinity of tacrine, donepezil, galantamine, and rivastigmine binding was considered as positive control which is -9.0 kcal/mol, -7.3 kcal/mol, -8.3 kcal/mol and -6.4 kcal/mol. Value (kcal/mol) greater than that was considered as the target ligand. The primary screening was performed by compounds with greater binding affinity than tacrine, rivastigmine, donepezil, and galantamine. A total of 620 molecules have exhibited higher binding affinity than the control molecules (tacrine, donepezil, rivastigmine, and galantamine), including 18 phytochemicals sourced from the Dr. Dukes database (https://phytochem.nal.usda.gov/) (S4 Table).

### ADME profiling of screened phytochemicals

The SwissADME (http://www.swissadme.ch/index.php) was utilized to examine the ADME profile and ability to traverse the blood-brain barrier for the selected 638 compounds. During this phase of the investigation, most of the chemicals did not meet the drug-likeness property that was assessed. Lipinski's rule states that, historically, 90% of orally absorbed drugs had fewer than 5 H-bond donors, less than 10 H-bond acceptors, molecular weight of less than 500 Daltons and XlogP values of less than 5 [25]. Due to their high solubility, many phytochemicals

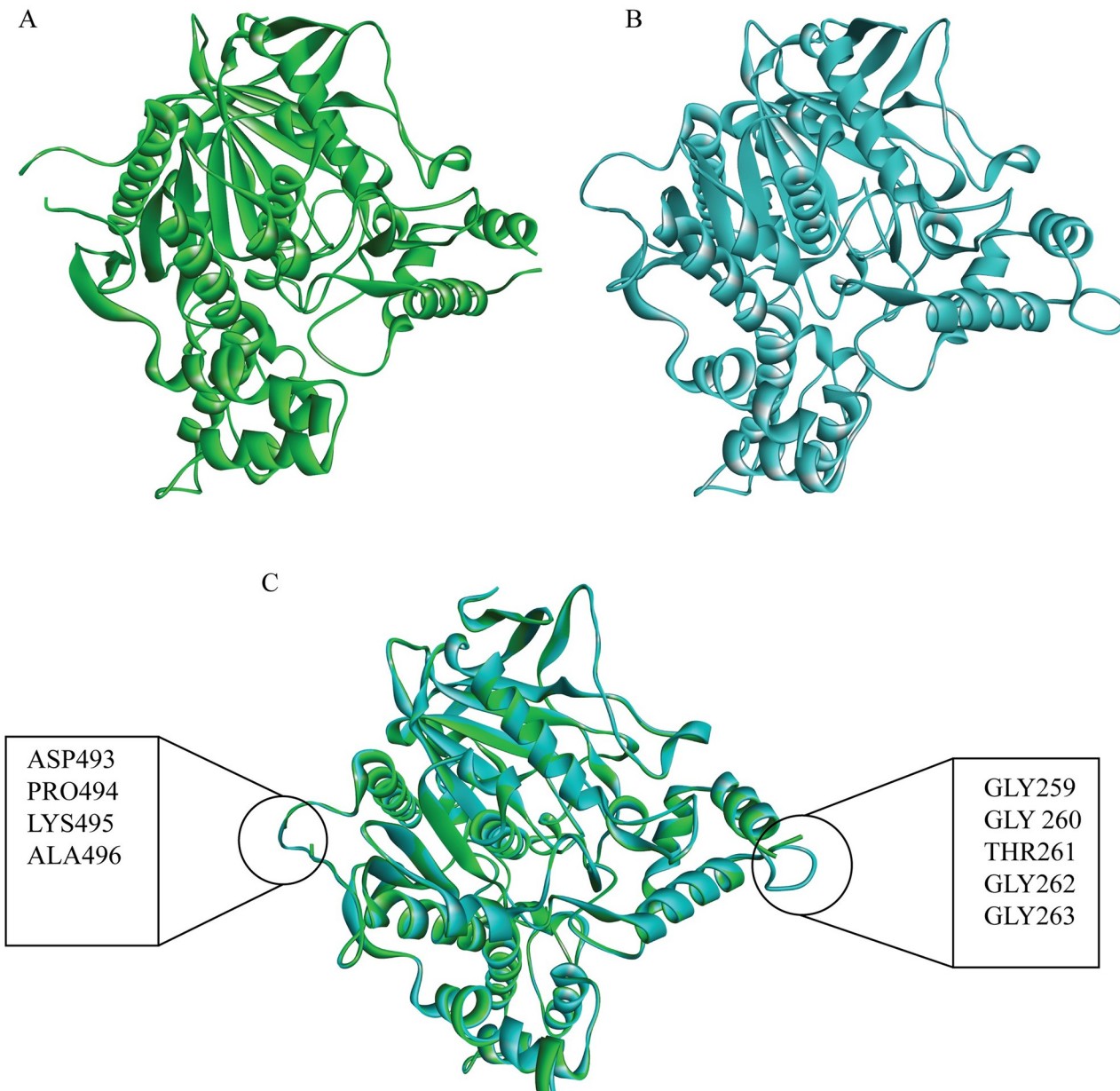

**Fig 1. Structure of AChE.** (A) The Crystal Structure of AChE Retrieved from RSCB-PDB, (B) the I-tasser predicted structure, and (C) the merged RCSB-PDB and predicted structures. The resolved missing residues and the conservation of the protein structure compared to its actual PDB sequence are shown.

may struggle to penetrate the blood-brain barrier (BBB). Therefore, compounds with a blood-brain barrier permeability (BBB) equal to or higher than 0.477 (Log 3) were prioritized for analysis as potentially potent BBB-permeable candidates. Additionally, high gastrointestinal (GI) absorption was assessed. A comprehensive analysis of the ADME (absorption, distribution, metabolism, and excretion) and docking results of Rivastigmine analogues (Table 2), Tacrine analogues (Table 3), Galantamine analogues (Table 4), and phytochemicals (Table 5) were

**Table 2. The docking, redocking and ADME results of Rivastigmine's similar structure with CID and chemical name.**

| Sl no | CID | IUPAC Name | Binding Affinity PyRx | Redocking (Autodock vina) | Hdock docking | BBB | Rules 5 violation | GI absorption | Leadlikeness violations |
|---|---|---|---|---|---|---|---|---|---|
| 1 | 77991 | Rivastigmine | -6.56±0.19 | -6.36±0.11 | -166.81 | 0.508 | 0 | High | 0 |
| 2 | 70266158 | [2-[1-(azetidin-1-yl)ethyl]phenyl] N,N-dimethylcarbamate | -7.16±0.70 | -7.07±0.37 | -176.94 | 0.564 | 0 | High | 2 |
| 3 | 66717459 | [3-[(1S)-1-(dimethylamino)ethyl]-2-tritiophenyl] N-ethyl-N-methylcarbamate | -8.1±0.5 | -7.9±0 | -191.39 | 0.506 | 0 | High | 1 |
| 4 | 42604975 | [3-[(1S)-1-[methyl-[(1S)-1-phenylethyl]amino]ethyl]phenyl] N-ethyl-N-methylcarbamate | -8.03±0.05 | -7.9±0.3 | -219.43 | 0.501 | 0 | High | 2 |
| 5 | 129309692 | [3-[1-[[(1S)-1-cyclohexa-1,3-dien-1-ylethyl]-methylamino]ethyl]phenyl] N-ethyl-N-methylcarbamate | -7.9±0 | -7.53±0.01 | -200.05 | 0.502 | 0 | High | 2 |
| 6 | 68377091 | [3-[(1S)-1-(dimethylamino)ethyl]phenyl] N-ethynyl-N-[(2R)-1-phenylpropan-2-yl]carbamate | -7.67±0.01 | -8.2±0 | -223.99 | 0.516 | 0 | High | 2 |
| 7 | 144066490 | [3-[1-(dimethylamino)ethyl]phenyl] N-methyl-N-[(2R)-1-phenylpropan-2-yl]carbamate | -7.2±0.75 | -7.77±0.20 | -194.04 | 0.506 | 0 | High | 3 |
| 8 | 10989924 | [3-(1-methylpiperidin-2-yl)phenyl] N,N-diethylcarbamate | -7.6±0 | -7.53±0.60 | -187.46 | 0.528 | 0 | High | 2 |
| 9 | 11359764 | [3-[(1S)-1-[methyl(trideuterio(113C)methyl)amino]ethyl]phenyl] N-methyl-N-(1,1,2,2,2-pentadeuterio(213C)ethyl)carbamate | -7.43±0.25 | -7.3±0.01 | -193.22 | 0.506 | 0 | High | 0 |
| 10 | 46898202 | [3-(1-piperidin-1-ylethyl)phenyl] N,N-diethylcarbamate | -7.53333 | N/A | -195.06 | 0.506 | 0 | High | 2 |
| 11 | 149047000 | [3-[1-(dimethylamino)cyclopropyl]phenyl] N-ethyl-N-methylcarbamate | -7.3±0.25 | -7.36±0.25 | -191.1 | 0.505 | 0 | High | 2 |
| 12 | 144474639 | [3-[(1S)-1-[[(1S)-1-cyclohexa-2,4-dien-1-ylethyl]-methylamino]ethyl]phenyl] N-ethyl-N-methylcarbamate | -7.16±0.40 | -7.13±0.01 | -212.22 | 0.501 | 0 | High | 0 |
| 13 | 21767521 | 7-[1-(dimethylamino)ethyl]-3-methyl-5,6-dihydro-4H-1,3-benzoxazocin-2-one | -7.46±0.05 | -6.96±0.20 | -179.55 | 0.555 | 0 | High | 2 |
| 14 | 21767510 | 6-[1-(dimethylamino)ethyl]-3-methyl-4,5-dihydro-1,3-benzoxazepin-2-one | -7.4±0 | -6.43±0.45 | -172.4 | 0.546 | 0 | High | 0 |
| 15 | 25204947 | [3-[(1S)-1-(dimethylamino)ethyl]phenyl] N-methyl-N-[(2S)-1-phenylpropan-2-yl]carbamate | -7.36±0.05 | -8±0.3 | -196.88 | 0.506 | 0 | High | 1 |
| 16 | 72816136 | [3-[1-(dimethylamino)ethyl]phenyl] N-methyl-N-(1-phenylpropan-2-yl)carbamate | -7.36±0.50 | -7.73±0.05 | -194.04 | 0.506 | 0 | High | 2 |
| 17 | 13955119 | [2-[1-(dimethylamino)ethyl]phenyl] N,N-dimethylcarbamate | -6.93±0.55 | -6.7±0.85 | -159.43 | 0.478 | 0 | High | 2 |
| 18 | 141557115 | [3-[1-(dimethylamino)pentyl]phenyl] acetate | -6.93±0.30 | -6.43±0.60 | -175.65 | 0.566 | 0 | High | 1 |
| 19 | 21767515 | 9-[1-(dimethylamino)ethyl]-3-methyl-5,6-dihydro-4H-1,3-benzoxazocin-2-one | -7±0.3 | -6.73±0.45 | -181.7 | 0.547 | 0 | High | 1 |
| 20 | 21767496 | 5-[1-(dimethylamino)ethyl]-3-methyl-4H-1,3-benzoxazin-2-one | -6.9±0.4 | -6.9±0.45 | -165.9 | 0.540 | 0 | High | 1 |
| 21 | 10935608 | [2-(1-piperidin-1-ylethyl)phenyl] N,N-diethylcarbamate | -7.2±0 | -7.36±.55 | -192 | 0.520 | 0 | High | 0 |
| 22 | 10924256 | [3-(piperidin-1-ylmethyl)phenyl] N,N-diethylcarbamate | -7.36±.35 | -7.06±0.14 | -178.19 | 0.517 | 0 | High | 1 |
| 23 | 144474633 | [3-[(2S)-1-(dimethylamino)propan-2-yl]phenyl] N-ethyl-N-methylcarbamate | -6.9±0.15 | -6.66±0.299 | -182.68 | 0.503 | 0 | High | 0 |
| 24 | 25230721 | [3-[(1S)-1,2,2,2-tetradeuterio-1-(dimethylamino)ethyl]phenyl] N-ethyl-N-methylcarbamate | -6.9±0.25 | -6.66±0.35 | -165.91 | 0.509 | 0 | High | 1 |
| 25 | 51037855 | [3-[(1S)-1,2,2,2-tetradeuterio-1-(dimethylamino)(213C)ethyl]phenyl] N-ethyl-N-methylcarbamate | -6.9±0.25 | -6.73±0.05 | -165.82 | 0.508 | 0 | High | 0 |

(*Continued*)

**Table 2.** (Continued)

| Sl no | CID | IUPAC Name | Binding Affinity PyRx | Redocking (Autodock vina) | Hdock docking | BBB | Rules 5 violation | GI absorption | Leadlikeness violations |
|---|---|---|---|---|---|---|---|---|---|
| 26 | 51038065 | [3-[(1S)-1-[methyl(trideuterio(113C)methyl)amino] ethyl]phenyl] N-ethyl-N-methylcarbamate | -6.76±.24 | -6.36±0.05 | -165.38 | 0.508 | 0 | High | 0 |
| 27 | 21767507 | [3-[(1S)-1-[methyl(trideuterio(113C)methyl)amino] ethyl]phenyl] N-methyl-N-(1,1,2,2,2-pentadeuterio (213C)ethyl)carbamate | -7.43±0.25 | -7.36±0.05 | -180.75 | 0.508 | 0 | High | 0 |
| 28 | 9823072 | [3-[(1S)-1-(dimethylamino)ethyl]-2-tritiophenyl] N-ethyl-N-methylcarbamate | -6.83±0.2 | -7.03±0.18 | -175.19 | 0.497 | 0 | High | 0 |
| 29 | 53705187 | [2-[[ethyl(methyl)amino]methyl]phenyl] N,N-dimethylcarbamate | -6.5± 0.35 | -6.36±0.35 | -153.59 | 0.493 | 0 | High | 1 |
| 30 | 97357026 | [3-[(1R)-1-(dimethylamino)ethyl]phenyl] N,N-diethylcarbamate | -6.63±.30 | -6.6±0.25 | -186.94 | 0.517 | 0 | High | 0 |
| 31 | 11066683 | [3-(1-piperidin-1-ylethyl)phenyl] N,N-diethylcarbamate | -6.96±0.19 | -7.33±0.15 | -187.83 | 0.493 | 0 | High | 1 |
| 32 | 25230725 | [3-[(1S)-1-[bis(trideuteriomethyl)amino]-1,2,2,2-tetradeuterioethyl]-2,4,5,6-tetradeuteriophenyl] N-(1,1,2,2,2-pentadeuterioethyl)-N-(trideuteriomethyl)carbamate | -7±0.15 | -7±0.45 | -165.02 | 0.517 | 0 | High | 0 |
| 33 | 144198864 | (1S)-1-(3-methoxyphenyl)-N,N-dimethylpropan-1-amine | -6.33±0.18 | -6±0.25 | -145.74 | 0.508 | 0 | High | 0 |
| 34 | 67474850 | [3-[(1S)-1-(dimethylamino)ethyl]-4-fluorophenyl] N-ethyl-N-methylcarbamate | -6.33±.44 | -6.1±0.55 | -166.31 | 0.764 | 0 | High | 0 |
| 35 | 10999871 | [3-(piperidin-1-ylmethyl)phenyl] N,N-dimethylcarbamate | -6.8±0.15 | -6.83±0.53 | -181.29 | 0.545 | 0 | High | 1 |
| 36 | 10586926 | [3-[(1S)-1-(dimethylamino)ethyl]-2-tritiophenyl] N-ethyl-N-methylcarbamate | -6.83±0.20 | -6.76±.60 | -166.78 | 0.533 | 0 | High | 0 |
| 37 | 71316042 | [3-(1-piperidin-1-ylethyl)phenyl] N,N-diethylcarbamate | -6.56±0.19 | -6.4±0.01 | -176 | 0.508 | 0 | High | 0 |
| 38 | 745584 | [2-[(dimethylamino)methyl]phenyl] N,N-dimethylcarbamate | -6.13±0.34 | -6.5±0.2 | -158.37 | 0.493 | 0 | High | 0 |
| 39 | 25230720 | [2-deuterio-3-[(1S)-1-[dideuteriomethyl(methyl) amino]ethyl]phenyl] N-ethyl-N-methylcarbamate | -6.6±0 | -6.67±0.455 | -163.7 | 0.532 | 0 | High | 0 |
| 40 | 25230723 | [3-[(1S)-1-(dimethylamino)ethyl]phenyl] N-ethyl-N-(trideuteriomethyl)carbamate | -6.43±0.35 | -6.93±0.50 | -165.17 | 0.508 | 0 | High | 1 |
| 41 | 25230724 | [3-[(1S)-1-(dimethylamino)ethyl]phenyl] N-methyl-N-(1,1,2,2-pentadeuterioethyl)carbamate | -6.6±0 | -6.56±0.51 | -165.17 | 0.508 | 0 | High | 0 |
| 42 | 51037853 | [3-[(1S)-1,2,2,2-tetradeuterio-1-(dimethylamino) (113C)ethyl]phenyl] N-ethyl-N-methylcarbamate | -6.73±0.18 | -7.033±0.49 | -165.82 | 0.508 | 0 | High | 0 |
| 43 | 51038067 | [3-[(1S)-1-[methyl(trideuterio(113C)methyl)amino] ethyl]phenyl] N-methyl-N-(1,1,2,2,2-pentadeuterio (213C)ethyl)carbamate | -6.73±0.29 | -6.5±0.35 | -189.38 | 0.508 | 0 | High | 0 |
| 44 | 77991 | [3-(1-piperidin-1-ylethyl)phenyl] N,N-diethylcarbamate | -6.66±0.01 | -6.36±0.15 | -176.57 | 0.508 | 0 | High | 0 |
| 45 | 92044359 | [3-[(1R)-1-[bis(trideuteriomethyl)amino]ethyl]phenyl] N-ethyl-N-methylcarbamate | -6.56±0.19 | -6.36±0.11 | -166.81 | 0.508 | 0 | High | 0 |

performed. Analogues of donepezil cannot fulfil the criteria of the ADME profile and are eliminated for further study. These tables provide valuable insights into the compounds' pharmacokinetic properties and their potential interactions with target proteins. A total of 89 compounds along with phytochemicals were found to possess the properties (S5 Table).

**Table 3. The Docking and redocking results of tacrine's similar structures with CID and chemical name.**

| Sl no | CID | IUPAC Name | Affinity Pyrx Kcal/mol | Redocking Autodock Kcal/mol | Hdock docking | BBB | Rules 5 violation | GI absorption | Leadlikeness violations |
|---|---|---|---|---|---|---|---|---|---|
| 1 | 1935 | Tacrine | -8.86±0.01 | -8.33±0.50 | -150.2 | 0.316 | 1 | High | 1 |
| 2 | 18403988 | 2-naphthalen-2-ylquinolin-4-amine | -10.23±0.05 | -10.1±0.65 | -202.24 | 0.565 | 0 | High | 2 |
| 3 | 149800 | N-benzylacridin-9-amine | -10.03±0.10 | -9.16±0.23 | -207.62 | 0.625 | 0 | High | 1 |
| 4 | 402658 | 12-azatetracyclo[9.8.0.02,7.013,18]nonadeca-1(19),2,4,6,11,13,15,17-octaen-19-amine | -9.9±0.051 | -8.63±0.37 | -189.28 | 0.54 | 0 | High | 2 |
| 5 | 54474520 | 3-[2-(7-fluoroquinolin-2-yl)ethenyl]aniline | -9.3±0.7 | -8.7±1.75 | -200.72 | 0.596 | 0 | High | 2 |
| 6 | 3438772 | 2-phenyl-4-pyrrolidin-1-ylquinoline | -9.7±0 | -8.7±1.05 | -200.55 | 0.559 | 0 | High | 2 |
| 7 | 18934490 | N-phenylacridin-1-amine | -9.56±.20 | -8.26±0.84 | -209.77 | 0.485 | 0 | High | 3 |
| 8 | 11492743 | 4-fluoro-2-(6-fluoro-4-methylquinolin-2-yl)aniline | -9.4±0.3 | -0.73±0.35 | -183.8 | 0.602 | 0 | High | 2 |
| 9 | 69799851 | 4-Amino-2-styrylquinoline | -9.53±0.06 | -9.067±0.9 | -190.86 | 0.577 | 0 | High | 1 |
| 10 | 129829335 | 10-sulfidoacridin-10-ium | -9.2±0.25 | -8.6±1.2 | -143.64 | 0.708 | 0 | High | 0 |
| 11 | 164587579 | 2-benzyl-6-fluoroquinolin-4-amine | -8.63333 | N/A | | 0.692 | 0 | High | 2 |
| 12 | 130408026 | 2-(7-fluoro-2-phenylquinolin-3-yl)ethanamine | -8.63±1.14 | -8.43±1.11 | -189.83 | 0.533 | 0 | High | 2 |
| 13 | 22395290 | 2-[(E)-2-phenylethenyl]quinolin-4-amine | -9.46±0.049 | -3±0.30 | -190.91 | 0.521 | 0 | High | 0 |
| 14 | 69799851 | 2-(2-phenylethenyl)quinolin-4-amine | -9.5±0 | -8.33±1.3 | -204.78 | 0.521 | 0 | High | 2 |
| 15 | 696663 | 12-azatetracyclo[9.8.0.02,7.013,18]nonadeca-1(19),2,4,6,11,13,15,17-octaen-19-amine | -9.5±0 | -8.8±0.85 | -185.5 | 0.495 | 0 | High | 0 |
| 16 | 402666 | 19-azatetracyclo[9.8.0.02,7.013,18]nonadeca-1(19),2,4,6,11,13,15,17-octaen-12-amine | -9.16±.20 | -8.83±1.59 | -191.76 | 0.483 | 0 | High | 1 |
| 17 | 10587156 | 6-fluoro-2-(2-fluorophenyl)quinolin-4-amine | -9.4±0 | -9.2±0.6 | -204.53 | 0.692 | 0 | High | 2 |
| 18 | 1504001 | 2-phenyl-4-piperidin-1-ylquinoline | -9.23±0.048 | -8.96±0.64 | -195.06 | 0.535 | 0 | High | 2 |
| 19 | 164587580 | 2-(2-fluorophenyl)quinolin-4-amine | -9.26±0.049 | -8.9±0.15 | -189.2 | 0.662 | 0 | High | 1 |
| 20 | 60598 | 9-(4-methylpiperidin-1-yl)-1,2,3,4-tetrahydroacridine | -8.96±0.1 | -9.46±.20 | -178.61 | 0.596 | 0 | High | 1 |
| 21 | 4452632 | 3-quinolin-2-ylaniline | -9.53±0.44 | -8.3±1.00 | -181.89 | 0.506 | 0 | High | 1 |
| 22 | 7742109 | (NZ)-N-(1-phenyl-2-quinolin-2-ylethylidene)hydroxylamine | -9.3±0 | -8.9±0.94 | -163.54 | 0.487 | 0 | High | 0 |
| 23 | 12102730 | 2,4-dimethylbenzo[h]quinolin-10-amine | -9.26±0.049 | -9.5±0 | -155.41 | 0.48 | 0 | High | 1 |
| 24 | 21998 | 10-methylacridin-10-ium-9-amine | -9.2±0 | -8.13±.133 | -178.3 | 0.71 | 0 | High | 0 |
| 25 | 45599224 | 12-azatetracyclo[9.8.0.02,7.013,18]nonadeca-1(19),2,4,6,11,13,15,17-octaen-19-amine | -9.2±0 | -8.26±1.15 | -184.9 | 0.653 | 0 | High | 1 |
| 26 | 45599463 | 5,7-difluoro-2-phenylquinolin-4-amine | -9.16±0.048 | -8.4±0.7 | -204.25 | 0.637 | 0 | High | 0 |
| 27 | 22334541 | N-(3-fluorophenyl)-2,3-dihydro-1H-cyclopenta[b]quinolin-9-amine | -9.2±0 | -7.13±0.94 | -189.01 | 0.635 | 0 | High | 0 |
| 28 | 11737199 | 2-(2-fluorophenyl)quinolin-4-amine | -9.2±0 | -8.73±1.44 | -182.28 | 0.583 | 0 | High | 0 |
| 29 | 55045454 | 6-methyl-2-phenylquinolin-4-amine | -9.1±0 | -9.2±0.3 | -158.98 | 0.484 | 0 | High | 0 |
| 30 | 31633 | 10-methylacridin-10-ium-3-amine | -9.16±0.048 | -9.93±0.15 | -186.82 | 0.71 | 0 | High | 1 |
| 31 | 45599470 | 7,8-difluoro-2-phenylquinolin-4-amine | -8.56±.71 | -7.43±1.52 | -182.25 | 0.701 | 0 | High | 0 |
| 32 | 45599222 | 6-fluoro-2-phenylquinolin-4-amine | -9.1±0 | -8.9±1.5 | -190.43 | 0.662 | 0 | High | 1 |
| 33 | 21828278 | 2,6-diphenylpyridin-4-amine | -9.06±0.15 | -7.43±0.048 | -174.1 | 0.613 | 0 | High | 0 |
| 34 | 21639083 | 12-azatetracyclo[9.8.0.02,7.013,18]nonadeca-1(19),2,4,6,11,13,15,17-octaen-19-amine | -8.96±0.149 | -8±0.7 | -196.44 | 0.607 | 0 | High | 0 |
| 35 | 43419931 | N-[(4-fluorophenyl)methyl]-2-methylquinolin-4-amine | -9.03±0.20 | -8.83±0.63 | -191.29 | 0.545 | 0 | High | 0 |
| 36 | 129641425 | 2-(2-phenylethenyl)quinolin-3-amine | -9.1±0 | -7.13±1.16 | -167.55 | 0.521 | 0 | High | 1 |
| 37 | 12394207 | 2-phenyl-4-piperidin-1-ylquinoline | -9.13±0.49 | -8.4±0.15 | -192.01 | 0.518 | 0 | High | 0 |
| 38 | 10980245 | 2-(2-fluorophenyl)quinolin-4-amine | -8.86±0.01 | -8.3±0.30 | -150.2 | 0.506 | 0 | High | 0 |

**Table 4. The Docking and redocking results of galantamine similar structure with CID and chemical name.**

| Sl no | CID (galantamine similar structures) | IUPAC Name | Affinity Pyrx Kcal/mol | Redocking Autodock Kcal/mol | Hdock docking | BBB | Rules 5 violation | GI Absorption | Leadlikeness violations |
|---|---|---|---|---|---|---|---|---|---|
| 1 | 9651 | Galantamine | -7.83±0.33 | -8.16±0.40 | -195.26 | -0.08 | 0 | High | 0 |
| 2 | 91042094 | 9-methoxy-4-prop-2-enyl-11-oxa-4-azatetracyclo[8.6.1.01,12.06,17] heptadeca-6(17),7,9,15-tetraene | -8.56±0.48 | -7.7±1.25 | -213.78 | 0.48 | 0 | High | 1 |
| 3 | 20706288 | 4,14-dimethyl-11-oxa-4 azatetracyclo [8.7.1.01,12.06,18]octadeca-6 (18),7,9,15-tetraen-9-ol | -8.43±0.48 | -8.6±0.5 | -194.87 | 0.59 | 0 | High | 0 |

## Computational molecular docking with AutoDock and Hdock

Outperforming control compounds tacrine, donepezil, galantamine, and rivastigmine, 89 identified molecules exhibit enhanced binding affinity in molecular docking via AutoDock Vina-1.5.7 and Hdock. These findings suggest their potential as promising acetylcholinesterase inhibitors, warranting further investigation, this study establishes a benchmark for assessing the comparative efficacy of the identified molecules with the positive control. The docking and redocking outcomes for the remaining compounds are comprehensively presented in the accompanying tables, encapsulating a comprehensive overview of their binding characteristics for further analytical consideration. This nuanced evaluation contributes to the burgeoning discourse surrounding potential therapeutic candidates for the development of novel acetyl-cholinesterase inhibitors [26].

The binding affinities of rivastigmine analogue compounds, which exhibit both blood-brain barrier (BBB) permeability and favourable drug-likeness characteristics, were further investigated (Table 2). Notably, three rivastigmine analogues, such as 10989924 ([3-(1-methyl-piperidin-2-yl)phenyl] N,N-diethylcarbamate), 74817986 ([3-[1-[methyl(1-phenylethyl) amino]ethyl]phenyl] N-ethyl-N-methylcarbamate) and 46898202 ([3-(1-piperidin-1-ylethyl) phenyl] N,N-diethylcarbamate), exhibited superior docking affinities as compared to rivastig-mine. This observation suggests a potential enhancement in the binding interactions of these molecules with the target receptor.

The binding affinities of tacrine and its structurally analogous exhibited the highest binding affinities in the entirety of the conducted docking study (Table 3). Notably, 2-naphthalen-2-ylquinolin-4-amine(18403988) emerges as the most promising candidate, displaying a sub-stantial binding affinity of -10.23±0.05 kcal/mol (PyRx), -10.1±0.65 kcal/mol (AutoDock) and -202.24 (Hdock). The overall binding affinities observed collectively underscore the potential of these compounds for further exploration and development. Conversely, the galantamine

**Table 5. The Docking results of phytochemicals with CID and chemical name.**

| Sl no | Ligand CID | IUPAC Name | Affinity Pyrx Kcal/mol | Redocking Autodock Kcal/mol | Hdock Docking | BBB | Rules 5 violation | GI Absorption | Leadlikeness violations |
|---|---|---|---|---|---|---|---|---|---|
| 1 | 2353 | Berberine | -8.63±0.78 | -9.2±0.4 | -244.81 | 0.198 | 0 | High | 1 |
| 2 | 5315472 | Bisdemethoxycurcumin | -8.86±1.1 | -8.83±0.65 | -232.61 | 0.398 | 0 | high | 0 |
| 3 | 6916252 | Huperzine B | -8.13±0.33 | -7.56±1.21 | -181.09 | 0.489 | 0 | High | 0 |
| 4 | 854026 | Huperzine A | -7.56±0.048 | -7.56±0.68 | -155.14 | 0.317 | 0 | High | 1 |
| 5 | 160512 | Ar-Turmerone | -7.63±0.14 | -6.83±0.48 | -155.14 | 0.105 | 1 | High | 2 |
| 6 | 1253 | (-)-Selagine | -7.26±0.28 | -6.66±0.31 | -174.59 | 0.512 | 0 | High | 1 |

Table 6. Docking site analysis for selected chemicals.

| Sl no | Ligand name | Complex | Pubchem CID | Pyrx Docking | Autodock docking | Interacting Residues |
|---|---|---|---|---|---|---|
| 1 | [3-(1-methylpiperidin-2-yl)phenyl] N,N-diethylcarbamate | Complex_1 | 10989924 | -7.6±0.0 | -7.53±0.6 | Tyr123, Tyr336, Tyr340, Phe337, Trp285 |
| 2 | 2-naphthalen-2-ylquinolin-4-amine | Complex_2 | 18403988 | 10.23±0.05 | -10.1±0.65 | Tyr123, Tyr285, Tyr340, His286, Asp73 |
| 3 | 4-Amino-2-styrylquinoline | Complex_3 | 69799851 | -9.5±0.0 | 8.33±1.3 | Tyr123, Phe337, Tyr336, Trp285, Trp85, Gly119, Gly120 |
| 4 | 9-methoxy-4-prop-2-enyl-11-oxa-4-azatetracyclo [8.6.1.01,12.06,17] heptadeca-6(17),7,9,15-tetraene | Complex_4 | 91042094 | -8.56±0.48 | -7.7±1.25 | Leu288, Leu75, Phe337, Phe296, Tyr340, Trp285 |
| 5 | Huperzine B | Complex_5 | 6916252 | -8.13±0.33 | -7.56±1.21 | Trp285, Tyr123, Tyr71, Leu71 |
| 6 | Bisdemethoxycurcumin | Complex_6 | 5315472 | -8.86±1.1 | -8.83±0.65 | Trp285, Tyr340, Trp85, Tyr71, His446 |
| 7 | Berberine | Complex_7 | 2353 | 8.63±0.78 | -9.2±0.4 | Tyr123, Tyr336, Tyr340, Phe337, Trp285, Ser292, His286 |
| 8 | Ar-Turmerone | Complex_8 | 160512 | -7.63±0.14 | -6.83±0.48 | Tyr123, Tyr336, Tyr340, Phe337, Trp285, Phe296, Leu288 |

similar structures present only two compounds, and among them 4,14-dimethyl-11-oxa-4 aza-tetracyclo [8.7.1.01,12.06,18]octadeca-6(18),7,9,15-tetraen-9-ol (20706288) was the best binding affinity with -8.43±0.48 Kcal/mol (PyRx), -8.6±0.5 Kcal/mol (Autodock) and -194.87 (Hdock), as the remaining analogues were judiciously excluded during primary virtual screening and ADME profiling (Table 4). This stringent selection process aims to ensure structural and pharmacokinetic viability, contributing to a refined pool of candidates with enhanced potential for subsequent stages of drug development.

Phytochemicals meeting the criteria of the blood-brain barrier (BBB) permeability and favourable drug-likeness were subjected to further investigation through molecular docking (Table 5). Among these, berberine exhibited a notable binding affinity of -8.63±0.78 kcal/mol (PyRx), -9.2±0.4 kcal/mol (autodock) and -244.81 (Hdock); huperzine B demonstrated -8.13 ±0.65 kcal/mol (PyRx), 7.56±1.21 kcal/mol and -181.09 (Hdock); bisdemethoxycurcumin revealed of -8.86±1.1 kcal/mol (PyRx), -8.83±0.65 kcal/mol (autodock) and 232.61 (Hdock); and Ar-Turmerone displayed a binding affinity o -7.63±0.14 kcal/mol (PyRx), -6.83±0.48 kcal/mol (autodock) and -155.81 (Hdock). These findings highlight the substantial potential of these phytochemicals as candidates for acetylcholinesterase inhibition.

## Docking site analysis

To conduct a more comprehensive investigation, a total of eight compounds (Table 6) have been chosen for a molecular dynamics (MD) simulation lasting 100 nanoseconds based on the docking analysis and ADME profiling. Utilizing BioVia Discovery Studio, it is feasible to visually observe the interaction between protein ligands and active site residues, as well as to overlay all proteins and ligands, based on their highest binding affinity and respective segments. The common residues involved in the positive controls tacrine, galantamine, rivastigmine, and donepezil are- Tyr340, Phe296, Trp285, Phe337, and Tyr123, and there was Tyr123 with a hydrogen bond and Trp285, Tyr340, and Phe296 with Pi-allyl interaction. However, the residues involved in the interaction and the binding sites exhibit similarities, as do the bonding characteristics. This suggests that the binding location and residues are congruent to those to which tacrine, donepezil rivastigmine galantamine bind.

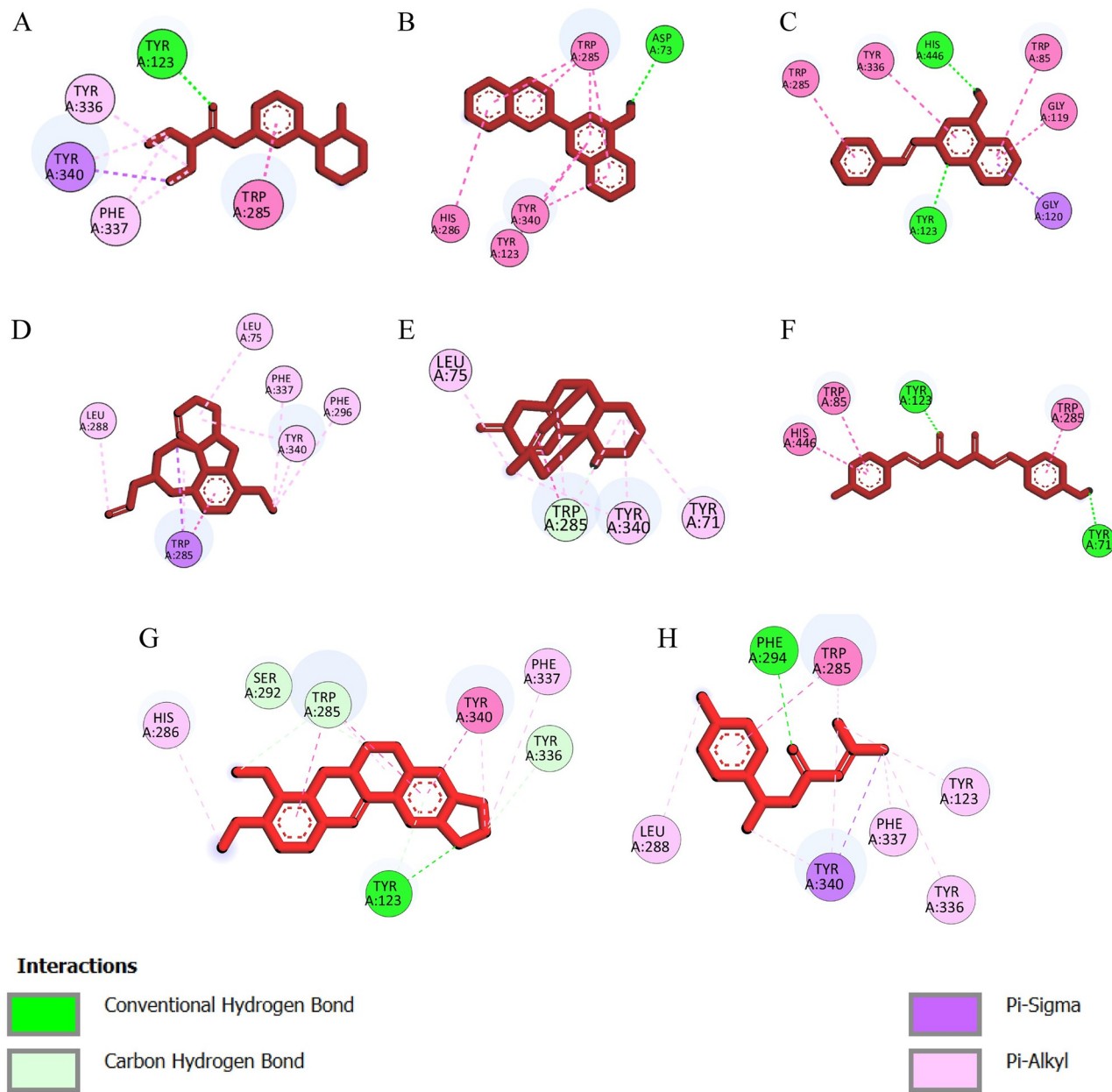

**Fig 2. A visual representation of Protein-ligand interaction.** The protein-ligand interaction of Complex_1 (A), Complex_2 (B), Complex_3 (C), Complex_4 (D), Complex_5 (E), Complex_6 (F), Complex_7 (G), and Complex_8 (H). All the interactions have common Tyr123 with a hydrogen bond and Trp85 with Pi-allyl interaction. The rest of the interactions have Pi-sigma with similar residues of the active side.

The 2D interaction analysis elucidates the nature of binding interactions (Fig 2), revealing the presence of pi-alkyl and pi-sigma interactions while notably excluding electrostatic bonds. Notably, TYR123 exhibits hydrogen bonding, and TRP285 displays pi-alkyl interaction across all complexes. These residue interactions demonstrate a consistent pattern, underscoring the reproducibility of specific binding motifs within the studied complexes.

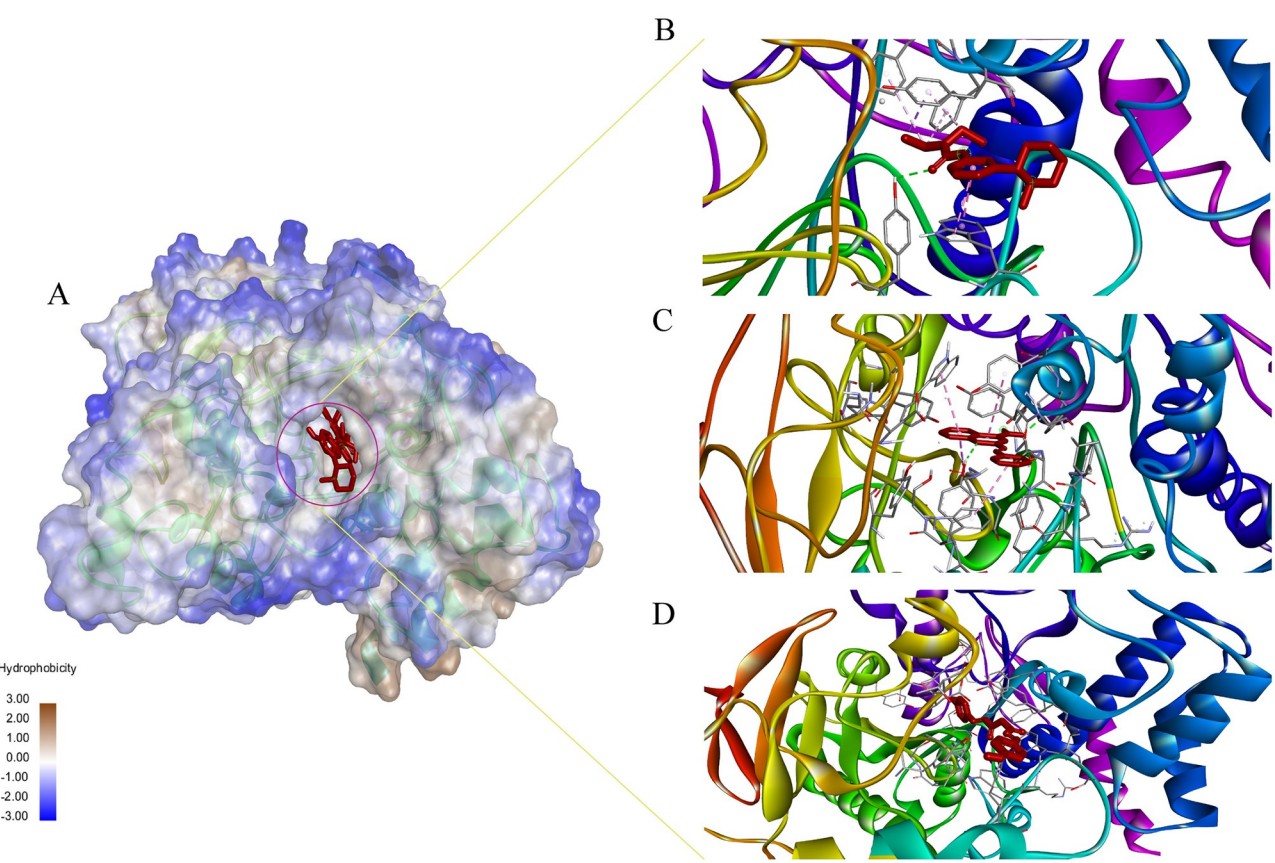

**Fig 3. A visual representation of the binding pocket and ligand interaction.** (A) The 3D Structure of protein-ligand complex and protein hydrophobicity mapping. Close view of Complex_1 (B), Complex_3 (C), Complex_6 (D). The protein pocket region is slightly bluish which indicates partially hydrophilic. All the ligands bind to the same side of the protein.

## Molecular dynamics simulation analysis

The simulation was performed in a Desmond environment. There were 8 compounds primarily selected for MD simulation in the Desmond simulation environment. The overall simulation results were interpreted in RMSD, RMSF, Ligand properties, DCCM, PCA and MM/GBSA analysis. The binding grooves (Fig 3A) of the examined chemicals were superimposed, revealing a remarkable degree of similarity in their spatial arrangements. Additionally, the residues involved in interactions exhibited striking congruence among the complex_1 (Fig 3B), complex_3 (Fig 3C), and complex_6 (Fig 3D). This congruency in binding grooves and interacting residues suggests a conserved mode of binding, reinforcing the likelihood of a shared molecular mechanism or target engagement.

The RMSD of Protein-ligand Complex figures have shown the Protein RMSD fit with ligand RMSD over a 100ns time scale. RMSD, which is the ligand insect in the protein RMSD line, is considered a good stability benchmark. Complex_1 Complex_3 and Complex_6 show better binding stability (Fig 4) and the other complexes couldn't show the stable binding affinity over the 100 ns time scale (S2 Fig). The good results were further confirmed by re-simulation of these 3 complexes using GROMACS software in the aspect of protein fit with ligand over the 100ns time scale (S3 Fig.) The Root Mean Square Fluctuation (RMSF) is a valuable tool for quantifying localized variations along the protein chain. Peaks on the plots represent

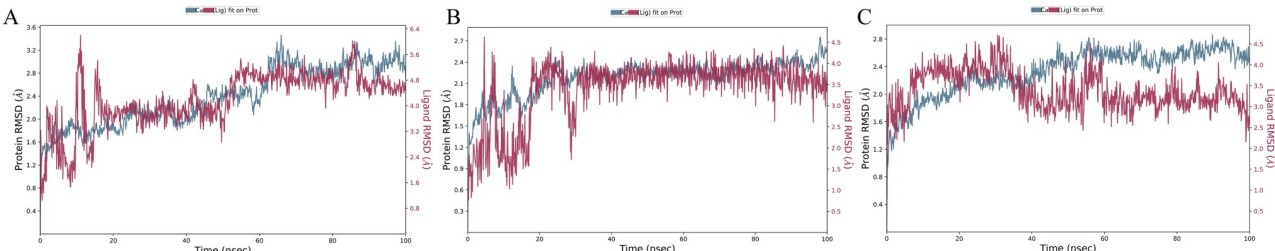

**Fig 4. A 100-nanosecond simulation is conducted to measure the root mean square deviation (RMSD) results of three complexes.** Complexes 1, 3, and 6 are subjected to be a better binding stability over the 100-nanosecond molecular dynamics simulation using the Desmond software. (A) RMSD of Complex_1, (B) RMSD of Complex_3, and (C) RMSD of Complex_6. The root means square deviation (RMSD) between the ligand and protein exhibits temporal constancy, thereby ensuring stability. Nevertheless, complex_1 and 3 demonstrate persistent stability, suggesting that the interaction between the protein and ligand remains intact throughout the entire duration. Complex_6 exhibits a deviation of 30ns, indicating inferior stability compared to the other 2 complexes. Nevertheless, the overall binding interaction is not significantly unfavourable, and further investigation is required for the other parameters.

regions of the protein that exhibit the highest degree of fluctuation throughout the simulation. It is commonly observed that the tails, specifically the N- and C-terminal, exhibit greater fluctuations compared to other regions of the protein. Secondary structure elements, such as alpha helices and beta strands, typically exhibit greater rigidity compared to the unstructured regions of the protein. As a result, they undergo less fluctuation than the loop regions (Fig 5), while the other five complexes RMSF have shown in S4 Fig. The RMSF results from the re-simulation using GROMACS software showed similar results (S5 Fig).

A ligand exhibiting a moderate degree of compactness, as measured by a moderate gyration value, could potentially achieve a harmonious equilibrium between sufficient molecular surface area (SASA) for interaction purposes and accessibility for binding. The combination of moderate gyration and a larger molecular surface area may provide numerous binding interaction sites, whereas a moderate SASA may indicate a stable structure with restricted solvent exposure (Fig 6).

The gyration results indicate that Complex_1 and Complex_3 are located within a range of 3.5–4.00 Armstrong, while Complex_6 is situated between 5.0–5.5 Å (Fig 6A). A higher value of the radius of gyration indicates a greater dispersion of atoms and a longer molecule. This metric quantifies the degree of elongation of a ligand and is equal to its primary moment of inertia. The SASA analysis reveals superior ligand characteristics, specifically in Complex_3

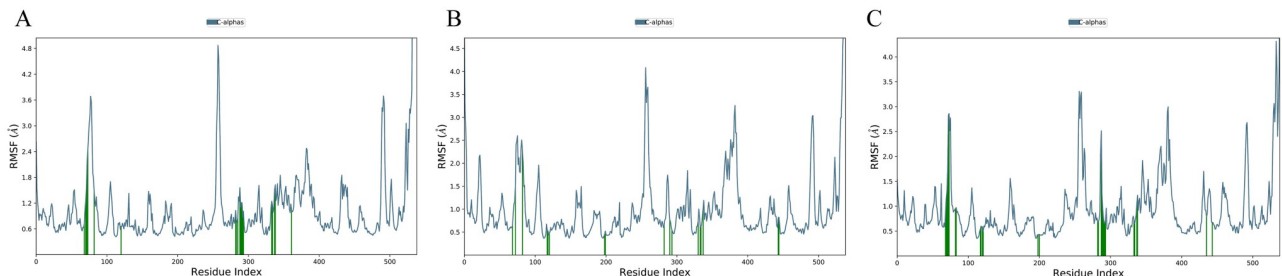

**Fig 5. The root means square fluctuation (RMSF) of all the simulation complexes over a 100-nanosecond simulation.** (A) Root Mean Square Fluctuation (RMSF) of Complex_1, (B) RMSF of Complex_3, and (C) RMSF of Complex_6. The interpretation of the results is justified. Several significant fluctuations. The fluctuation primarily arises when the ligand interacts with the protein residues. Complex_1 exhibits three significant fluctuations on the green vertical bar, which signify the contact between the ligand molecule and the protein. Complex_3 and Complex_6 exhibit significant temporal fluctuations. The overall comparison reveals significant fluctuations, although they do not exceed 4.8 AÅ.

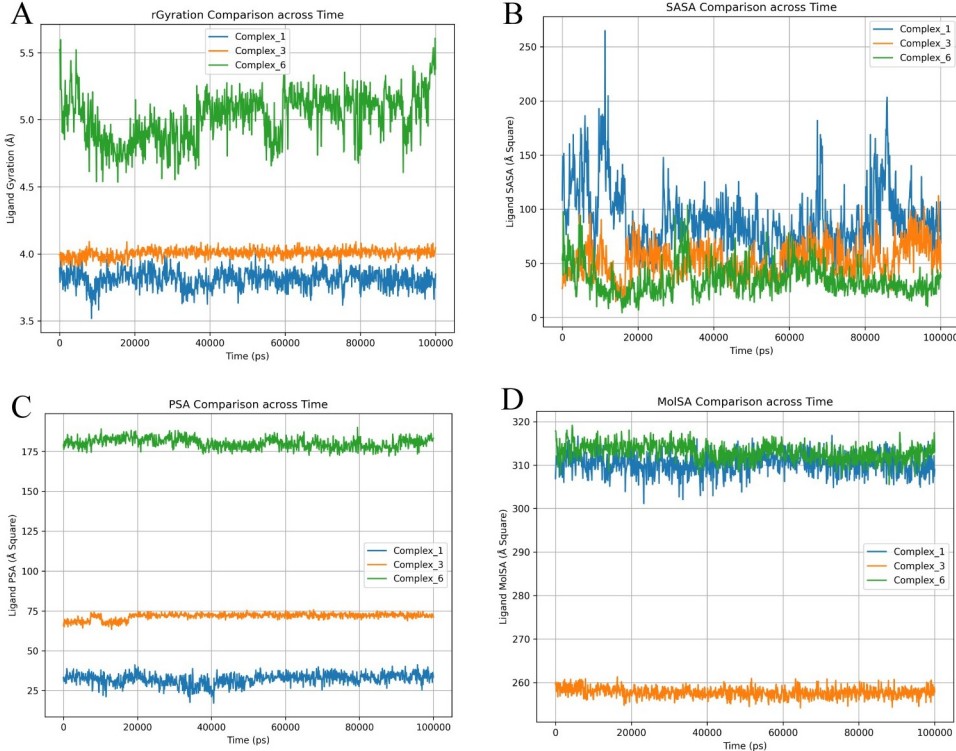

**Fig 6. A 100ns simulation of ligand properties of all the complexes.** (A) Ligand Gyration, (B) Ligand SASA, (C) Ligand Polar Surface Area (PSA), and (D) Molecular Surface Area (MolSA). Values of complex_1, complex_3, and complex_6 are represented with blue, orange, and green colours, respectively.

and Complex_6, with a surface area ranging from 50 to 100 Armstrong square units (Fig 6B). Reduced solvent-accessible surface area (SASA) leads to increased binding stability. The polar surface area and the molecular surface area exhibit significant differences. Complex_1 exhibits lower levels of PSA and higher levels of MolSA, whereas Complex_6 displays higher levels of both PSA and MolSA (Fig 6C and 6D). Complex_6 exhibits reduced levels of PSA and MolSA. Elevated PSA levels can potentially impact binding employing electrostatic interactions. A greater MolSA value signifies an increased number of sites available for interacting with other molecules or receptors. The ligand SASA, ligand Gyration, protein SASA, and protein gyration from the re-simulation using GROMACS software showed a similar pattern (S6 Fig.).

## PCA analysis

Principal Component Analysis (PCA) is a mathematical technique that identifies the most significant components in a dataset by analyzing the covariance or correlation matrix. In the context of protein analysis, PCA utilizes atomic coordinates to define the protein's available degrees of freedom (DOF). The result of those three results PCAs has been performed (Fig 7). PCA analysis of each of the component percentages indicates each of the parameters, PC1 might indicate how strongly the ligand binds to the protein, PC2 could represent something like the flexibility of the protein-ligand complex and PC3 might capture variations in the shape complementarity between the protein and ligand.

The highest percentage of variance explained is indicated by the Single Component with the Highest Variance (PC1), as determined by the PCA analysis. Complex_1 PCA yields the

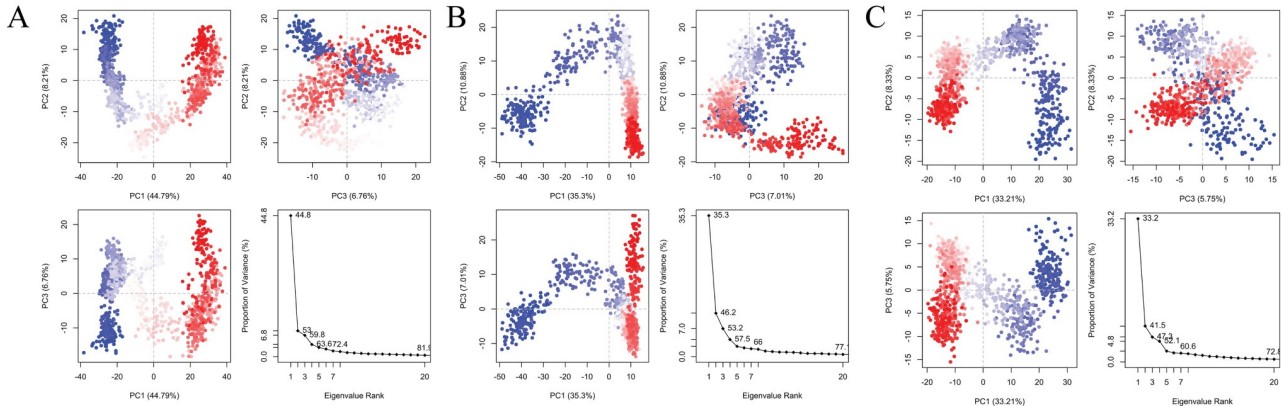

**Fig 7. PCA analysis of three complexes.** The PCA of Complex_1 (A), complex_2 (B), and complex_3 (C). The White dot here mentions the transition state of protein-ligand simulation confirmation, the blue dot with a scattered indicates energetically unstable conformational states and the red dots indicate the stable conformational state.

most favourable outcomes, followed by complex_3 and complex_6. By considering the amalgamation of constituents that capture substantial variation in contrast to the summaries of 46.18% and 41.54% for both complexes, Complex_1 exhibits a sum of 53% (Table 7). It exhibits improved variances. Complex 1 exhibits superior performance in both analyses, whether a singular component with the highest variance is considered or a collection of components that collectively account for a substantial proportion of the data's variance is considered [27].

## DCCM analysis

The DCCM analysis method was applied in a novel way to assist in the identification of potential protein domains. During the implementation of this novel approach, multiple DCCM maps (Fig 8) were computed, each utilizing a distinct coordinate reference frame to determine the boundaries of protein domains and the constituents of protein domain residues[28].

## MM/GBSA analysis

The binding free energies predicted by MM/GBSA for Complexes 1, 3, and 6 show a strong correlation with the calculated values. However, the strength of MM/PBSA and MM/GBSA lies in their ability to dissect the obtained binding free energies into specific components, such as the contributions from van der Waals interactions and hydrogen bonding from the solvent phase (Table 8). In assessing the overall protein complexes, it becomes evident that the unfavourable contribution primarily stems from covalent binding across all complexes suggesting that there is no favorable covalent interaction which this protein-ligand complexes. Notably, both Complex 1 and Complex 6 exhibit favorable outcomes in terms of Coulombic

**Table 7. Different PCA components chart of each of the complexes.**

| Complex | PCA Components | | |
|---|---|---|---|
| | PC1 (%) | PC2 (%) | PC3 (%) |
| Complex_1 | 44.7 | 8.21 | 6.76 |
| Complex_3 | 35.3 | 10.88 | 7.01 |
| Complex_6 | 33.21 | 8.33 | 5.75 |

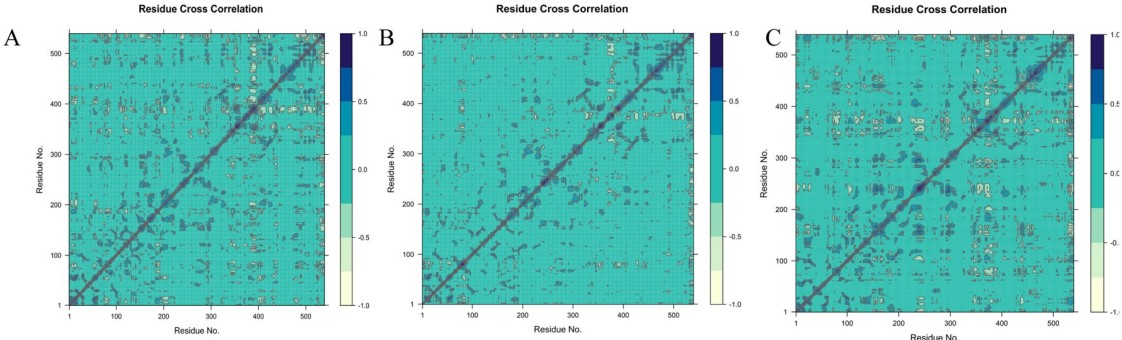

**Fig 8. The cross-correlation map of the C α atom pairs within the monomers of AChE is analyzed for dynamics.** The DCCM of Complex_1 (A), complex_2 (B), and complex_3 (C). The correlation coefficient (C ij) was represented using various colours. The values of Cij, ranging from 0 to 1, indicate positive correlations. Positive correlations indicate that these pairs of atoms tend to move in similar directions or have comparable behaviors during the simulation. On the other hand, negative correlations are represented by Cij values ranging from -1 to 0. Negative correlations indicate that these pairs of atoms tend to migrate in opposite directions or have contrasting behaviors during the simulation.

interactions, whereas Complex 3 demonstrates an unfavorable trend in this regard. The Hydrogen bond plays a crucial role in competitive inhibitory mechanisms. The greater HBond can seen in complex_3 which is -2.67913 kcal/mol (S6 Table).

## Discussion

The therapeutic intervention of Alzheimer's disease (AD) using AChEi has been demonstrated by a wide range of plant-based compounds [29]. Given the absence of reliable, efficient, and secure inhibitors, investigating structurally similar compounds could be a promising field for researchers to explore [30]. In this study, we analyzed the chemical structures of tacrine, done-pezil, galantamine, and rivastigmine to identify potential alternative drugs that are safer [31]. Computer aid drug design (CADD) methodologies have been discovered to expand the repositories of chemical compounds for the identification of potential inhibitors. The assessment of the binding affinity between a protein and a vast collection of ligands is frequently accomplished through the application of molecular docking techniques [32]. The molecules within the applicability domain of the constructed-in-silico model were screened to assess their drug-likeness and ADME properties. Drug likeness provides a highly valuable criterion for determining the minimum requirements that a compound must meet to be considered suitable for drug development [33]. This criterion helps in the objective selection of new drug candidates that have desirable bioavailability [34].

Molecular docking is a highly effective approach in CADD that utilizes specific algorithms to determine the affinity scores based on the positioning of ligands within the binding pocket of a target. In molecular docking, the lowest docking score corresponds to the highest affinity, indicating that the complex remains in contact for a longer period with good stability [35, 36].

**Table 8. Calculated binding free energy of protein for complex_1, complex_3 and complex_6 with its component contributions (all the units are in kcal/mol).**

| Sl no | Compound | ΔGbind (Kcal/mol) | ΔGbind Coulomb | ΔGbind Covalent | ΔGbind Vander | ΔGbind HBond | ΔGbind Lipophilic |
|-------|-----------|-------------------|----------------|-----------------|---------------|--------------|-------------------|
| 1 | Complex_1 | -47.21025 | -60.08772 | 2.78367 | -29.40811 | -0.14593 | -34.87072 |
| 2 | Complex_3 | -67.26226 | 9.254793 | 6.30836 | -34.90804 | -2.67913 | -29.17265 |
| 3 | Complex_6 | -49.89042 | -17.04911 | 5.36441 | -38.68284 | -0.55515 | -32.40361 |

Rigorously examine the protein-ligand binding to identify compounds with higher binding affinity and potentially improved hydrogen bonding characteristics [37]. The analysis of the docking results confirmed the binding of the final three compounds, including [3-(10methyl-piperidin-2-yl)phenyl]. The residues Tyr123, Tyr336, Tyr340, Phe337, and Trp285 are involved in the interaction with N,N-diethyl carbamate. Specifically, Compound 3, identified as 4-amino-2-styrylquinoline, interacts with the residues Tyr 123, Phe 337, Tyr 336, Trp 285, Trp 85, Gly 119, and Gly 120. Conversely, Compound 6, known as Bisdemethoxycurcumin, binds to the residues Trp 285, Tyr 123, Trp 85, Tyr 71, and His 446.

Molecular dynamics simulations demonstrated stable interactions between specific ligands and the AChE binding site. Notably, compounds like [3-(1-methylpiperidin-2-yl)phenyl] N, N-diethylcarbamate, 4-Amino-2-styrylquinoline and Bisdemethoxycurcumin displayed consistent and favourable interactions throughout the simulation period. Such stability suggests a potential for these compounds to serve as stable and effective inhibitors. The RMSD and RMSF values of these complexes remained quite stable throughout the simulation. Specifically, the complex involving 4-Amino-2-styrylquinoline exhibited stability with a constant value over time. Similarly, [3-(1-methylpiperidin-2-yl)phenyl] N,N-diethylcarbamate also demonstrated stability during the simulation. Although the RMSD of Bisdemethoxycurcumin deviated, indicating a slight variation in the protein-ligand fit, the overall stability remained satisfactory. PCA and DCCM analysis of those three compounds were performed. Principal Component Analysis (PCA) in molecular dynamics studies elucidates key factors influencing protein-ligand interactions. PC1 signifies binding strength, PC2 reflects protein-ligand complex flexibility, and PC3 captures shape complementarity. Higher PC1 scores denote stronger interactions, while elevated PC2 scores suggest increased complex flexibility. Enhanced PC3 scores indicate superior geometric fit between protein and ligand [27]. Complex_1, comprising [3-(1-methylpiperidin-2-yl) phenyl] N, N-diethyl carbamate, binds with AChE and demonstrates superior performance in PC analysis. Additionally, Compounds 3 (4-Amino-2-styrylquinoline) and 6 (Bisdemethoxycurcumin) exhibit promising results in PCA. Conversely, the DCCM analysis of compound 1 reveals a positive correlation among the protein-protein residues throughout the simulation, alongside stable correlations with certain compounds exhibiting both positive and negative associations [38]. The MMGBSA suggests that none of these complexes could be able to bind with the protein covalently which can suggest the drug doses and period. The hydrogen bond is also an important parameter for protein-ligand competitive inhibitory mechanisms. Complex_3 serves as a potential candidate for the AChE inhibitor. The simulation results it depict complex_3 serves as potent inhibitory properties against AChE

Exploring the potential of computationally screened compounds in comparison to established drugs for Alzheimer's disease shows a promising direction for future research [31]. Experimental validation using *in vitro* and *in vivo* studies is essential to confirm the effectiveness and safety characteristics of these identified compounds. Recognizing the constraints of the computational approach is crucial, including the inherent approximations in modelling, the possibility of false positives, and the requirement for experimental verification. The intricate characteristics of AD pathophysiology pose difficulties in identifying specific inhibitors that efficiently target the progression of the disease [39]. The combination of computational screening and molecular dynamics simulations provides an initial yet insightful view of potential inhibitors for AD [40]. The identified compounds show potential as candidates for further investigation and confirmation in preclinical and clinical studies. Nevertheless, the practical application of these compounds as effective treatments necessitates thorough experimental verification [41].

## Conclusion

The treatment of Alzheimer's disease through acetylcholinesterase inhibitors has been show-cased by various plant-derived compounds. Considering the scarcity of dependable, effective, and safe inhibitors, exploring compounds with comparable structures holds promise as a potential avenue for investigation. In this study, we performed a virtual screening to discover new cholinesterase inhibitors from similar structures of already approved drugs and plant compounds that interact with cholinesterase. Docking and molecular simulation tools were employed to investigate the significance of binding interactions of potentially new molecules for Alzheimer's disease treatment. The comparative analysis of molecular dynamics simulation data generated from two distinct software platforms elucidates a more nuanced understanding of the stability dynamics inherent in protein-ligand interactions. Additionally, the utilization of molecular mechanics generalized born surface area scoring across various parameters provides valuable insights that complement and potentially corroborate the hypothesized mechanisms.

## Supporting information

**S1 Fig. The Ramachandran plot.** The Ramachandran plot shows the statistical distribution of the combinations of the backbone dihedral angles φ and ψ. In theory, the allowed regions of the Ramachandran plot show which values of the Phi/Psi angles are possible for an amino acid, X, in an ala-X-ala tripeptide (Wiltgen, 2019). The Ramachandran plot analysis of protein AChE showed high conformational quality, with no outliers identified. All 537 residues (100%) were in acceptable regions (>99.8%), with 96.6% (519/537) falling within favoured regions (>98%).
(TIF)

**S2 Fig. A 100-nanosecond simulation is conducted to measure the root mean square deviation (RMSD).** Results of rest 5 complexes. Complexes 2, 4, 5, 7 and 8 are subjected to a 100-nanosecond molecular dynamics simulation using the Desmond software. A) RMSD of Complex_2, B) RMSD of Complex_4, C) RMSD of Complex_5, D) RMSD of Complex_7, E) RMSD of Complex_8. The root means square deviation (RMSD) between the ligand and protein exhibits temporal constancy, thereby can't able tov ensure stability. Nevertheless, complex_4 demonstrated persistent stability over 90 ns simulation but rest of 10 ns it deviated, and ligand don't fit to the protein, suggesting that the interaction between the protein and ligand remains intact throughout the 90 ns duration but ultimate result is not good all over the time. The rest of the complexes don't show any stable interaction throughout the simulation. It may cause of ion imbalance of the binding pore. Overall binding interaction is not favorable, and further investigation is required for the other parameters.
(TIF)

**S3 Fig. Root mean square deviation (RMSD) profile of selected compound.** Cross-validation of the stability by another simulation machine GROMACS. In the graph, green, and red colour represented protein carbon alpha chains, ligand respectively. A) Complex_1 showed steady protein RMSD, and ligands overlapped only for a short time. B) Complex_3 displayed increasing RMSD and deemed well synchronized with the ligand RMSD. C) Complex_6 demonstrates overlapping RMSD of protein and ligand till 60 ns afterward ligand RMSD goes higher. Overall, RMSD of the complexes ensures stability and compactness although further validation may need. Nevertheless, the overall binding interaction is not significantly unfavourable, and further investigation is required for the other parameters.
(TIF)

**S4 Fig. The root means square fluctuation (RMSF) of all the simulation complexes over a 100-nanosecond simulation.** A- Root Mean Square Fluctuation (RMSF) of Complex_2, B- RMSF of Complex_4, C- RMSF of Complex_5, D- RMSF of Complex_7, E- RMSF of Complex_8. The interpretation of the results is justified by Several significant fluctuations. The fluctuation primarily arises when the ligand interacts with the protein residues. Complex_7 has shown two major fluctuations as 180 residues and 395 residues. Overall, others exhibit significant fluctuations on the green vertical bar, which signify the hydrogen bond contact between the ligand molecule and the protein.
(TIF)

**S5 Fig. The root means square fluctuation (RMSF) of all the simulation complexes over a 100-nanosecond simulation.** A- Root Mean Square Fluctuation (RMSF) of Complex_1, B- RMSF of Complex_3, C- RMSF of Complex_6. The interpretation of the results is justified. Several significant fluctuations. The fluctuation primarily arises when the ligand interacts with the protein residues. Complex_1 exhibits three significant fluctuations, which signify the contact between the ligand molecule and the protein. Complex_3 and Complex_6 exhibit significant temporal fluctuations. The overall comparison reveals significant fluctuations, although they do not exceed 0.4 nm.
(TIF)

**S6 Fig. A 100 ns simulation of ligand properties for all the complexes.** The green, orange, and blue colours represent Complex_1, Complex_3, and Complex_6, respectively. A) The SASA value of the ligand across 100 ns displayed lower solvent access. B) The gyration of the ligand results indicates that Complex_1 and Complex_3 are located within a range of 0.40 nm, while Complex_6 averages 0.55 nm. Additionally, Complex_6 showed irregularity in the radius of gyration, notably around 20 ns, indicating changes in atom dispersion around an axis.
(TIF)

**S1 Table. The active side residues.**
(DOCX)

**S2 Table. The sequence alignment of 3D predicted structures.**
(DOCX)

**S3 Table. The geometrical analysis of 3D predicted protein.**
(DOCX)

**S4 Table. ADME analysis of screened chemical compounds.**
(DOCX)

**S5 Table. Molecular docking study of primary screening by pyrx.**
(DOCX)

**S6 Table. Predicted binding free energies for complexes 1, 3, and 6.**
(XLSX)

## Acknowledgments

The authors acknowledge the logistic support and laboratory facilities of the Department of Biochemistry and Molecular Biology, Shahjalal University of Science and Technology, Sylhet, Bangladesh.

## Author Contributions

**Conceptualization:** Rashid Taqui, Ajit Ghosh.

**Data curation:** Mahir Azmal, Md. Sahadot Hossen.

**Formal analysis:** Mahir Azmal, Md. Sahadot Hossen, Md. Naimul Haque Shohan, Abbeha Malik.

**Funding acquisition:** Ajit Ghosh.

**Investigation:** Mahir Azmal, Md. Sahadot Hossen, Md. Naimul Haque Shohan, Abbeha Malik.

**Methodology:** Mahir Azmal, Abbeha Malik, Ajit Ghosh.

**Software:** Abbeha Malik.

**Supervision:** Abbeha Malik, Ajit Ghosh.

**Validation:** Rashid Taqui, Abbeha Malik.

**Visualization:** Ajit Ghosh.

**Writing – original draft:** Mahir Azmal.

**Writing – review & editing:** Ajit Ghosh.

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
