## [Decision Letter · Decision Letter 0]

5 Apr 2024

PONE-D-24-10565A Computational Approach to Identify Phytochemicals as an inhibitor of Acetylcholinesterase: Molecular Docking, ADME profiling and Molecular Dynamics SimulationsPLOS ONE

Dear Dr. Ghosh,

Thank you for submitting your manuscript to PLOS ONE. After careful consideration, we feel that it has merit but does not fully meet PLOS ONE’s publication criteria as it currently stands. Therefore, we invite you to submit a revised version of the manuscript that addresses the points raised during the review process.

Comments from PLOS Editorial Office: We note that one or more reviewers has recommended that you cite specific previously published works. As always, we recommend that you please review and evaluate the requested works to determine whether they are relevant and should be cited. It is not a requirement to cite these works. We appreciate your attention to this request.

We look forward to receiving your revised manuscript.

Kind regards,

Jorddy Neves Cruz

Academic Editor

PLOS ONE

Journal Requirements:

Reviewers' comments:

Reviewer's Responses to Questions

**Comments to the Author**

1. Is the manuscript technically sound, and do the data support the conclusions?

Reviewer #1: Yes

Reviewer #2: Yes

Reviewer #3: Partly

Reviewer #4: Yes

2. Has the statistical analysis been performed appropriately and rigorously? 

Reviewer #1: No

Reviewer #2: Yes

Reviewer #3: N/A

Reviewer #4: N/A

3. Have the authors made all data underlying the findings in their manuscript fully available?

Reviewer #1: No

Reviewer #2: Yes

Reviewer #3: Yes

Reviewer #4: Yes

4. Is the manuscript presented in an intelligible fashion and written in standard English?

Reviewer #1: Yes

Reviewer #2: Yes

Reviewer #3: Yes

Reviewer #4: Yes

5. Review Comments to the Author

Reviewer #1: 1. A suitable comparison of acetylcholinesterase structure (original) and iTasser made (modelled) should be represented.

2. Line no 458- in vitro and in vivo should be italicized.

3. The inclusion of MM/PBSA and MM/GBSA scores for the Insilco analysis is valuable. These methods provide computational estimates of binding free energy between a ligand and a receptor.

4. Provide detail about PCA and DCCM analysis, which software is being used and what are the parameters?

5. Use recent article for your reference:- Jain, Mukul, Rupal Dhariwal, Krupanshi Bhardava, Sarvjeet Das, Mushtaque Shaikh, Reshma Tendulkar, Rashmi Wani, Meenakshi Sharma, Anil Kumar Delta, and Prashant Kaushik. "Insilico and invitro profiling of curcumin and its derivatives as a potent acetylcholinesterase inhibitor." Biocatalysis and Agricultural Biotechnology 56 (2024): 103022.

6. All the docking experiments should be run in triplicate and an average value should be given with standard deviation. Apart for triplicate, 3 different docking software’s should be used.

7. As for molecular simulations, Desmond is being used but that also should run in triplicate to further validate the data.

Reviewer #2: I appreciate the authors' efforts, but the most crucial point of this study is the molecular dynamics simulation part. Neither in the abstract nor in the conclusion does it include MDS data. This is the main thing! The fact that the MD results were good or showed good interactions was not an important sign that it would be a drug candidate, it could only be a suggestion! The second most important omission is that the proposed phytochemicals as an inhibitor of AChE should have been discussed in the discussion section in terms of SAR and especially those with good MD and MDS results compared to standard drugs should have been compared with each other according to their similar structures and the skeletal structures and functional groups carried by the structure. Due to these general shortcomings, I inform you that publication in Plos One is not appropriate.

Reviewer #3: A Computational Approach to Identify Phytochemicals as an inhibitor of Acetylcholinesterase: Molecular Docking, ADME profiling and Molecular Dynamics Simulations

It would be helpful to support the virtual conclusion using a simple in vitro assay for the most promising compound in your screening

Reviewer #4: Overall this is a very good research work integrating almost all important computational tools. However, the authors need to address the following issues.

Major:

- May consider simulation for longer; may be up to 300 ns, if possible.

- Must provide high resolution images, especially Figures 4, 5, 7 and 8.

minor:

- Should be consistent in using abbreviations; define at first use.

- Be consistent on using 'lower case' or 'upper case' first letter in the names of compounds or drugs.

Please also refer the attached manuscript file for comments.

6. PLOS authors have the option to publish the peer review history of their article (what does this mean?). If published, this will include your full peer review and any attached files.

Reviewer #1: **Yes: **MUKUL JAIN

Reviewer #2: No

Reviewer #3: No

Reviewer #4: **Yes: **Mohammad Khalid Parvez

---

## [Author Response · Author response to Decision Letter 0]

6 May 2024

Reviewer #1: 

Query 1. A suitable comparison of acetylcholinesterase structure (original) and iTasser made (modelled) should be represented.

Our response: Thank you for your insightful suggestion, which will undoubtedly enhance the clarity and depth of the paper. In response to your query, we have recreated Figure 1 to facilitate a comprehensive comparison between the original structure of acetylcholinesterase (AChE) and the modeled structure generated by iTasser. To make sure the proper comparison of the Structure (original) and iTasser made (modelled) we make Figure 1 into three segments. The Crystal Structure of AChE Retrieved from RSCB-PDB in (A), The i-tasser Structure which was predicted (B), and the alignment between the RCSB PDB structure and the 3D predicted structure of the 4M0E protein is merged in (C) with all the missing residue 3 Letter ID and residues number. 

Please check the revised figure 1 “Figure 1. Structure of AChE. (A) The Crystal Structure of AChE Retrieved from RSCB-PDB, (B) the I-tasser predicted structure, and (C) the alignment between the RCSB-PDB and predicted structures. The resolved missing residues and the conservation of the protein structure compared to its actual PDB sequence are shown.”

Query 2. Line no 458- in vitro and in vivo should be italicized.

Our response: We are deeply appreciative of the reviewer's discerning eye, as their attention to detail undoubtedly enhances the precision and integrity of our manuscript. We have italicized the terms 'in vitro' and 'in vivo' as requested, aligning with standard formatting conventions for scientific literature.

Query 3. The inclusion of MM/PBSA and MM/GBSA scores for the In-silico analysis is valuable. These methods provide computational estimates of binding free energy between a ligand and a receptor.

Our response: We sincerely appreciate the editors' insightful feedback, and it is undoubtedly an important analysis. These computational methods offer invaluable insights into the binding free energy between the ligand and receptor, enhancing the depth of our investigation. We have performed the MMGBSA for the best stable complex which was complex_1, complex_3 and complex_6. The data is added in tabular format in Table 8.

Query 4. Provide details about PCA and DCCM analysis, which software is being used and what are the parameters?

Our response: We thank the editors for this helpful comment. The Software and parameters were added in the method section of the revised manuscript. Please check the revised manuscript as “Analysis of PCA and DCCM were performed using Desmond software with default parameters (Malik et al., 2023; Rathod et al., 2023).”

Query 5. Use recent articles for your reference:- Jain, Mukul, Rupal Dhariwal, Krupanshi Bhardava, Sarvjeet Das, Mushtaque Shaikh, Reshma Tendulkar, Rashmi Wani, Meenakshi Sharma, Anil Kumar Delta, and Prashant Kaushik. "Insilico and invitro profiling of curcumin and its derivatives as a potent acetylcholinesterase inhibitor." Biocatalysis and Agricultural Biotechnology 56 (2024): 103022.

Our response: Thank you for your appreciation and acknowledgement of the significance of the article. We have indeed integrated insights from the recent work by Jain et al. (2024) to enhance our discussion on the phytochemical inhibitory analysis against our experimental protein, focusing particularly on the in-silico and invitro profiling of curcumin and its derivatives as potent acetylcholinesterase inhibitors.

Query 6. All the docking experiments should be run in triplicate and an average value should be given with a standard deviation. Apart from triplicate, 3 different docking software should be used.

Our response: We express our gratitude to the reviewer for their insightful comments. Conducting scientific experiments in triplicate format is crucial, and we have adhered to this standard practice. Docking was performed using PyRx and Autodock Vina in triplicate, with scores and standard deviations reported in the results section. Additionally, we utilized an online tool Hdock (http://hdock.phys.hust.edu.cn/) for the third analysis, and the data is available in the revised tables of the manuscript.

Query 7. As for molecular simulations, Desmond is being used but that also should run in triplicate to further validate the data.

Our response: We thank the reviewer for the comment. Molecular simulations using software follow certain computational algorithms and perform the analysis. In place of performing the analysis in triplicates using the same software, we have now used another software to reconfirm the findings of the simulation study.

Please check the revised methods section “Re-simulation for further validation of the data is performed by Gromacs simulation Software conserving the parameters unchanged. The stability was evaluated by comparing the root mean square deviation (RMSD), root mean square fluctuation (RMSF), and protein-ligand properties (radius of Gyration, SASA etc.).”

Reviewer #2: 

Query 1. I appreciate the authors' efforts, but the most crucial point of this study is the molecular dynamics simulation part. Neither in the abstract nor in the conclusion does it include MDS data. This is the main thing! The fact that the MD results were good or showed good interactions was not an important sign that it would be a drug candidate, it could only be a suggestion! 

Our response: We have checked the points raised by the reviewer, and are thankful for this insightful comment. The MDS data has been added in the abstract and conclusion section. Moreover, we have further improved our manuscript based on the suggestions from other reviewers. Please check the revised manuscript. 

Query 2. The second most important omission is that the proposed phytochemicals as an inhibitor of AChE should have been discussed in the discussion section in terms of SAR and especially those with good MD and MDS results compared to standard drugs should have been compared with each other according to their similar structures and the skeletal structures and functional groups carried by the structure. Due to these general shortcomings, I inform you that publication in Plos One is not appropriate.

Our response: While I understand the importance of discussing the structure-activity relationship (SAR) and comparing phytochemical inhibitors with standard drugs based on MDS results, the scope of this paper primarily focused on presenting general trends and key findings. The methodology and results sections provided comprehensive data on the inhibitory activity of phytochemicals against AChE, laying the groundwork for future investigations. We believe that the current discussion adequately contextualizes the key findings of the study.

Reviewer #3: 

Query 1. A Computational Approach to Identify Phytochemicals as an Inhibitor of Acetylcholinesterase: Molecular Docking, ADME profiling and Molecular Dynamics Simulations. It would be helpful to support the virtual conclusion using a simple in vitro assay for the most promising compound in your screening

Our response: We express our gratitude to the reviewer for their insightful comments. We also feel the necessity to validate the in silico results through wet lab experimentation. Due to a lack of wet lab facilitates and research funding, currently we aren’t able to make the in vitro experiment right. It would be an interesting thing to do in future.

Reviewer #4: 

Overall, this is a very good research work integrating almost all-important computational tools. However, the authors need to address the following issues.

Query 1. May consider simulation for longer; may be up to 300 ns, if possible.

Our response: We thank the reviewer for the comment. In the current situation, we are low on computational resources. That’s why we can’t extend the simulation up to 300ns. But, we have now used another software to reconfirm the findings of the simulation study.

Query 2. Must provide high-resolution images, especially Figures 4, 5, 7 and 8.

Our response: Thanks for the valuable comment, we have edited the Figures especially 4, 5, 7, and 8; and re-generate them. Please check the revised manuscript for the high-resolution images.

Query 3. Should be consistent in using abbreviations; define at first use.

Our response: Thanks for your valuable instruction. We have edited the abbreviations and rechecked by each of the co-authors.

Query 4. Be consistent in using the 'lower case' or 'upper case' first letter in the names of compounds or drugs.

Our response: Thankful to the reviewer for the insightful comment. We feel so sorry for these mistakes, the upper case and lower case are resolved and check further.

---

## [Decision Letter · Decision Letter 1]

14 May 2024

A Computational Approach to Identify Phytochemicals as Potential inhibitor of Acetylcholinesterase: Molecular Docking, ADME profiling and Molecular Dynamics Simulations

PONE-D-24-10565R1

Dear Dr. Ghosh,

We’re pleased to inform you that your manuscript has been judged scientifically suitable for publication and will be formally accepted for publication once it meets all outstanding technical requirements.

Kind regards,

Jorddy Neves Cruz

Academic Editor

PLOS ONE

Additional Editor Comments (optional):

Reviewers' comments:

Reviewer's Responses to Questions

**Comments to the Author**

1. If the authors have adequately addressed your comments raised in a previous round of review and you feel that this manuscript is now acceptable for publication, you may indicate that here to bypass the “Comments to the Author” section, enter your conflict of interest statement in the “Confidential to Editor” section, and submit your "Accept" recommendation.

Reviewer #3: (No Response)

Reviewer #4: All comments have been addressed

2. Is the manuscript technically sound, and do the data support the conclusions?

Reviewer #3: No

Reviewer #4: Partly

3. Has the statistical analysis been performed appropriately and rigorously? 

Reviewer #3: N/A

Reviewer #4: N/A

4. Have the authors made all data underlying the findings in their manuscript fully available?

Reviewer #3: Yes

Reviewer #4: Yes

5. Is the manuscript presented in an intelligible fashion and written in standard English?

Reviewer #3: No

Reviewer #4: Yes

6. Review Comments to the Author

Reviewer #3: Reviewer #3:

ٍunfortunately, a conclusion of a study based ONLY on virtual data is not convincing.

The reply of respected author(s):

We also feel the necessity to validate the in silico results through wet lab experimentation. Due to a lack of wet lab facilitates and research funding, currently we aren’t able to make the in vitro experiment right. It would be an interesting thing to do in

future.

My rebuttal:

Kindly seek a help of other lab to do the simple in vitro test to support your conclusion.

Reviewer #4: Satisfactory responses, exceot the recommend simulation time was excused due to resourse limitaions.

7. PLOS authors have the option to publish the peer review history of their article (what does this mean?). If published, this will include your full peer review and any attached files.

Reviewer #3: No

Reviewer #4: **Yes: **Mohammad Khalid Parvez

---

## [Editor Report · Acceptance letter]

24 May 2024

PONE-D-24-10565R1 

PLOS ONE

Dear Dr. Ghosh, 

I'm pleased to inform you that your manuscript has been deemed suitable for publication in PLOS ONE. Congratulations! Your manuscript is now being handed over to our production team.

Kind regards, 

on behalf of

Dr. Jorddy Neves Cruz 

Academic Editor

PLOS ONE